pone.0254331

Witwatersrand, SOUTH AFRICA

**Data Availability Statement:** There are ethical
restrictions on sharing the data as stated by both
the MSF and South Sudan ERBs. Data cannot be

# Evaluation of a community-based HIV test and start program in a conflict affected rural area of Yambio County, South Sudan

**Cecilia Ferreyra**[1]*, **Laura Moretó-Planas**[1], **Fara Wagbo Temessadouno**[1,2],
**Beatriz Alonso**[3], **Buai Tut**[3], **Victoria Achut**[4], **Mohamed Eltom**[5], **Endashaw M. Aderie**[3],
**Vicente Descalzo-Jorro**[3]

1 Médecins sans Frontières, Barcelona, Spain, 2 Épicentre, Dakar, Senegal, 3 Médecins sans Frontières,
Juba, South Sudan, 4 Ministry of Health, Republic of South Sudan, HIV/AIDS/STI, Juba, South Sudan,
5 Médecins sans Frontières, Nairobi, Kenya

* ferreyracecilia@gmail.com

## Abstract

### Background

Antiretroviral therapy (ART) coverage in South Sudan is around 10%. Access to HIV care in
settings with low ART coverage or conflict affected is still low; innovative strategies are
needed to increase access and ensure continuation of ART during instability. A pilot HIV
test and start project was implemented in a conflict-affected area of South Sudan. In a retro-
spective analysis, we determined the feasibility and outcomes of this intervention.

### Methods

Programme data from July 2015 to June 2018 was analysed. The project involved five
mobile teams offering HIV counselling and testing (HCT) and same day ART initiation at
community level. Baseline and follow-up information on clinical, immunological and viral
load (VL) was routinely recorded, as well as treatment outcomes. A semi-qualitative study
was conducted to assess acceptability of the program among beneficiaries and community
members.

### Results

By June 2018, 14824 people received counselling and testing for HIV and 498 (3.4%) tested
positive. Out of those 395 (79.3%) started ART. A total of 72 ART patients were organized in
26 Community ART Groups (CAGs) and contingency plan was activated 9 times for 101
patients. Kaplan-Meier estimated retention in care (RIC) at 12 and 18 months was 80.6%
[95% CI: 75.9–84.5%] and 69.9% [95% CI: 64.4–74.8%] respectively. RIC was significantly
higher at 18 months in patients under community ART groups (CAGs) (90.9% versus 63.4%
p<0.001) when compared to patients on regular follow up. VL suppression at 12 months
was 90.3% and overall virological suppression reached 91.2%. A total of 279 persons were
interviewed about the MSF program perception and acceptance: 98% had heard about the

shared without requesting permission, as defined in the MOU. The data underlying the results presented in the study are available from MSF OCBA. (www.msf.es) by contacting the medical director: cristian.casademont@barcelona.msf.org.

**Funding:** The author(s) received no specific funding for this work.

**Competing interests:** The authors have declared that no competing interests exist.

programme and 84% found it beneficial for the community, 98% accepted to be tested and only 4% found disadvantages to the programme.

## Conclusions

Our study shows that HCT and early ART initiation in conflict affected populations can be provided with good program outcomes. RIC and virological suppression are comparable with facility-based HIV programs and to those in stable contexts. This model could be extrapolated to other similar contexts with low access to ART and where security situation is a concern.

## Introduction

According to UNAIDS, HIV prevalence in South Sudan was 2.5% in 2018 making it a generalized epidemic [1, 2] and ART coverage in the country ranges between 14 and 18% [3, 4]. An estimated 16000 new HIV infections happen every year and about 13000 die due to AIDS related illness. South Sudan is one of the 38 high priority countries for UNAIDS. In a recent report, AIDS related deaths have been recorded to be the second highest cause of mortality for internally displaced persons in Protection of Civilians (PoC) sites in UN bases, due to the lack of comprehensive HIV services [1]. Former Western Equatoria State had one of the highest HIV prevalence rates in the country: between 8 to 12%. MSF had been working in Yambio State Hospital (YSH) since 2006, offering access to health to a particularly exposed to violence population. Medical activities included care for Human African Trypanosomiasis (HAT), paediatric inpatient department (IPD), outpatient department (OPD) and Sexual and Reproductive health (SRH). Due to the high HIV prevalence reported in the area during 2012 MSF began supporting prevention from mother to child HIV transmission (PMTCT) and HIV care to adults and children. As of January 2015, more than 3000 patients were enrolled into HIV care at YSH and 1500 patients were started on ART. HIV services were centralized at hospital level and there were no primary health care centres, particularly in rural areas, providing access to HIV care. The MSF HIV program covered a total catchment population of 96112 in Yambio district.,

Access to HIV care in settings with low ART coverage or affected by conflict is far from the WHO recommended targets of 90% ART coverage [5–7]. The combination of a less visible epidemic profile, competing medical priorities, weak health services and lack of human resources pose considerable obstacles to HIV interventions [8]. Even in settings where HIV services are available, to some extent most of these contexts still fall behind in the implementation proven successful strategies of HIV care, such as task shifting and decentralization. Therefore bringing HIV test, treatment and adherence services closer to the patient's home, with simplified protocols, and tools which allow for comparable treatment outcomes to facility-based care, could be a more suitable strategy in areas with limited human resources or centralized HIV services [9, 10].

A conflict setting, acute or protracted, is an added challenge for the provision of HIV testing, treatment and adherence services. Other medical humanitarian priorities make HIV care to fall among the lowest of the priorities [11–14]. Little political engagement has been seen in the past few years to scale up HIV testing, treatment and adherence services in regions such as West and Central Africa (WCA) and other low prevalence settings. Some experiences have been reported about models of provision of HIV care in these settings, however this has not yet been translated into political and donor's commitment to close this gap [15–17].

Community or home based HIV counselling and testing (CB-HCT and HB-HCT) leads to access to antiretroviral therapy (ART) for infected people and access to prevention services for the uninfected [18]. CB-HCT and HB-HCT have already been implemented in Sub-Saharan African countries demonstrating varying levels of testing uptake and acceptability, however little evidence is available from conflict affected areas [9, 19–22]. These strategies bring HIV testing closer to the people's home reaching previously untested persons unaware of their HIV status. The strategy has been recommended by WHO for generalized HIV epidemics with linkage to prevention, care and treatment services [10].

Several studies, report high drop-out rates between HIV testing and ART initiation in referral clinics compared to community-based interventions [11, 18, 23]. Initiating ART at the same site and time of testing, with or without CD4 testing, would increase ART uptake, and consequently reduce HIV related morbidity, mortality, transmission and incidence [5, 24, 25]. Thus the benefit of ART initiation on the immune compromised patients in advanced stages of the disease and on those with higher CD4 cell count has already been documented with clear evidence of improved individual patient health and hence reducing HIV transmission in the community [25, 26].

The progress against HIV/AIDS over the past 15 years has inspired a global commitment to end the epidemic by 2030; despite the unprecedented efforts to increase the number of people being tested for HIV and started on antiretroviral treatment (ART) some areas of the world are still left behind [27]. Coverage in access to HIV testing and ART services has remained disproportionately low in vulnerable and marginalized populations.

This study describes program outcomes including retention in care and viral suppression of a community-based HIV test and start program in Yambio; a rural and conflict-affected population in South Sudan.

## Methodology

This pilot program was designed utilizing a two-phase observational study approach. The first phase consisted of information and communication campaign at community level after which community based mobile clinics provided HCT and ART initiation. The second phase consisted of monitoring retention in care and clinical and virological outcomes at 12 and 18 months. The project started in June 2015 and all patients were handed over to a local organization supported by South Sudan's Ministry of Health by end of June 2018. All patient data were collected at day 0 when HCT was offered and at each follow up encounter during the study period, paper forms were later on entered in Epidata database for data management. Data collected included sociodemographic characteristics as well as clinical and laboratory data. Data were exported to and analysed using STATA v.15. Continuous variables were summarized using mean and standard deviation or median and interquartile range as appropriate and were expressed as ordinal categories with frequencies reported with corresponding 95% confidence intervals. The main outcomes of interest were retention in care, viral suppression, LFU and death at 12 and 18 months of programme initiation. Kaplan Meir was used to determine the retention in care and Cox regression model to assess the factors associated with retention in care, and viral suppression. We used a stepwise backward regression technique at $p<0.2$ to include variables in risk factor assessment model. All variables with $p$-value$>0.2$ were sequentially removed in the process. A $p$-value$<0.05$ was used for inclusion of factors in the final model.

A semi-qualitative assessment to understand community perception and acceptability of the program was done at community level including participants as follows: a) people to whom HIV testing was offered and accepted, b) people who refused to be tested c) people who

tested HIV positive but refuse ART initiation d) people who tested HIV positive and started ART but stopped at some point e) people who completed follow up and f) people with negative HIV test results. The number of participants for each category of people was limited when we observed a saturation of the responses we received.

This study was approved by the MSF and the South Sudan Ministry of Health Ethics Review Boards

## Health promotion and community engagement (HPCE) implementation

Community Health Workers (CHWs) were identified and trained on HIV messaging and community-based HIV programs. An information and communication campaign adapted to the local community took place to inform about the project and its objectives. The contents of this campaign were discussed with community leaders and local population to better understand behaviours related with HIV transmission and perception. Community leaders were informed before starting the program to enable their engagement in the program.

Once the community was familiar with the team and the planned activities, door-to-door sensitization was carried out. Appointments for individual or smaller group visits were taken at the end of the common meetings. Radio messages and announcements, fliers, posters, t-shirts and loud-speaking microphones, were also used to reinforce direct mobilization.

The mobile teams were assigned to "drop-in centres", which usually were well-known places in the centre of the village, such as markets, churches, schools, etc. In order to increase the uptake of the services by the community, the program also included a "market-to-market" strategy where one mobile team moved from one market place to another on specific market days including weekends to offer HIV test and treatment.

### HCT and ART adherence counselling implementation

Every mobile clinic had 3 counsellors trained for HCT and ART adherence counselling. The patients attending the drop-in centre would first receive a health education session about HIV and ART provided by the CHWs. Afterwards, every client would receive individual pre-test counselling, referred to the mobile lab for HIV testing and go back to the counsellor for individual post-test counselling (Fig 1). Patients tested HIV positive would get baseline diagnostic tests and then addressed to a clinical officer for ART initiation on the same day. At every visit, all patients would receive ART adherence counselling. Patients with suspected ART failure would undergo enhanced adherence counselling during 3 months followed by viral load monitoring.

### Laboratory test

A trained laboratory assistant was part of the mobile teams who carried all the lab tests needed for that day. Serial HIV testing algorithm was performed using rapid diagnostic test (RDT) Determine® and Uni-Gold® with capillary blood obtained by finger prick (Fig 2). When results were discordant, tests were repeated on whole blood and, if still discordant, repeated after 2 weeks. Second discordant results were sent for DNA-PCR to Global Clinical and Viral Laboratory (GCVL), Durban, South Africa. All HIV patients underwent baseline CD4 cell count, viral load (VL), syphilis, creatinine and haemoglobin tests. CD4 cell count was performed every 6 months using point of care PIMA® machine. VL was analysed using dried-blood-spots (DBS) samples that were sent to GCVL South Africa for viral load determination at initiation and follow-up appointments (3, 6, 12 and 18 months). Patients with suspected

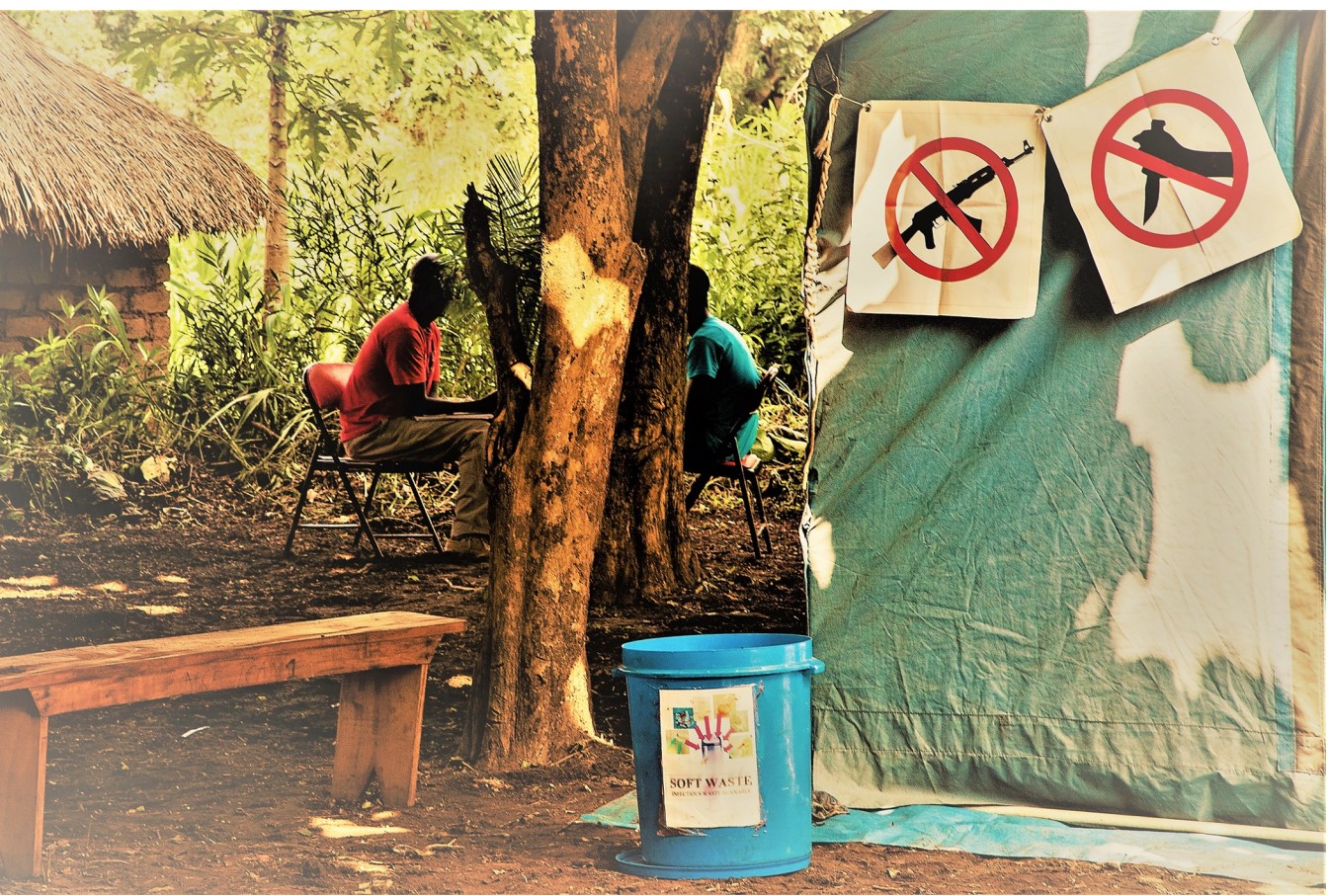

**Fig 1. Individual counselling session at community level.**

ART failure (viral load over 1000 copies/ml after 6 months on ART) had a new VL after 3 months of enhanced adherence counselling.

## ART initiation and follow-up

Clinical (WHO stage) and immunological (CD4 count) situation were assessed at the first visit and ART was initiated on the same day after obtaining informed consent, regardless of the CD4 count and WHO clinical staging (Test and start strategy).

At first consultation, contact information (address, telephone) was recorded for all patients. Follow-up consultations took place at 2 weeks, 1 month and 3 months interval afterwards during the subsequent visits. Drug side effects, adherence status, and development of immune reconstitution syndrome (IRS) were assessed using structured forms at those follow-up visits. Clinically ill or unstable patients would be referred to YSH and reason for referral recorded.

Contact information of patients not attending the consultation was shared with CHW who would then be in charge of tracing the client at community level. Lost to follow-up patients were defined as those patients who missed their appointment for 2 months.

## Community ART groups (CAGs)

CAGs are self-formed groups of stable patients on ART who take turns attending clinical assessment and monitoring tests at the drop-in centre, whilst collecting drugs for themselves

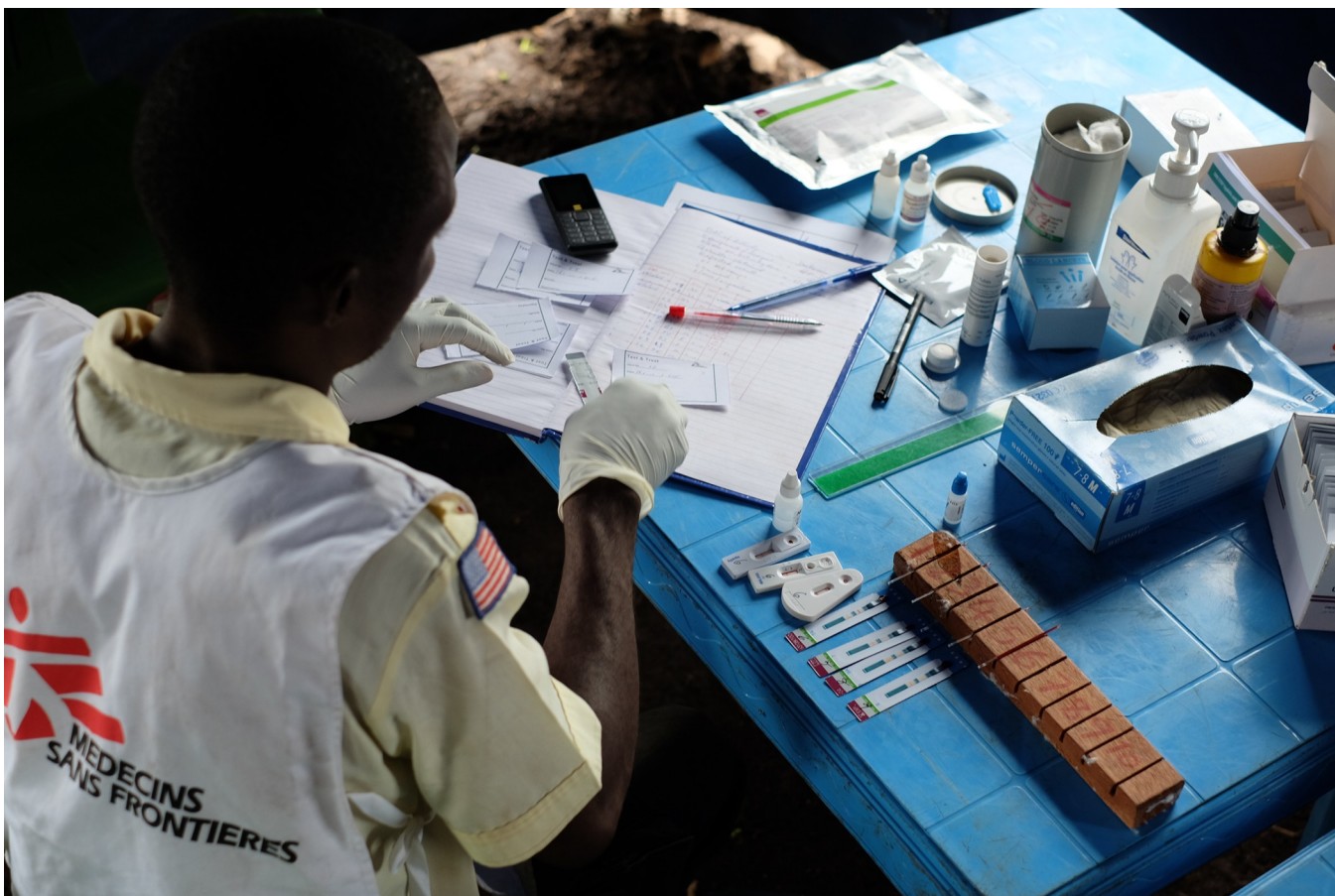

**Fig 2. Rapid diagnostic testing for HIV, CD4 and viral load at community level.**

and the other members of the group. In our program the groups were formed to support patients who lived in remote areas and to ensure no disruption of treatment in case of security incidents. Only patients on ART with stable clinical condition (no active TB or other co-infection), HIV viral suppression and good adherence were included. The groups consisted of 3 to 6 adult members from the same community who would visit drop-in centres in turns every 1 or 2 months intervals. Creation of CAGs required having at least one person being able to read and write, at least one person with phone and who could easily contact the rest of the group members. Initial trainings were provided to CAG members to reinforce confidentiality, clarify misconceptions and sensitize about technical aspects to fill the CAG monitor reports and registers.

## Contingency plan

Due to the unstable security context in the area, a contingency plan to ensure continuation of treatment in case of service disruption was put in place. Patients were counselled since the first visit about the risk of follow up disruption and how to ensure continuation of ART if that happened. All the mobile teams and CHWs were trained on how to implement this plan following different levels of insecurity. Three months' supply of ARVs and drugs for opportunistic infections were pre-packed for each patient in a "run-away bag" ready to be delivered to the patient by the CHW on a short notice (Fig 3).

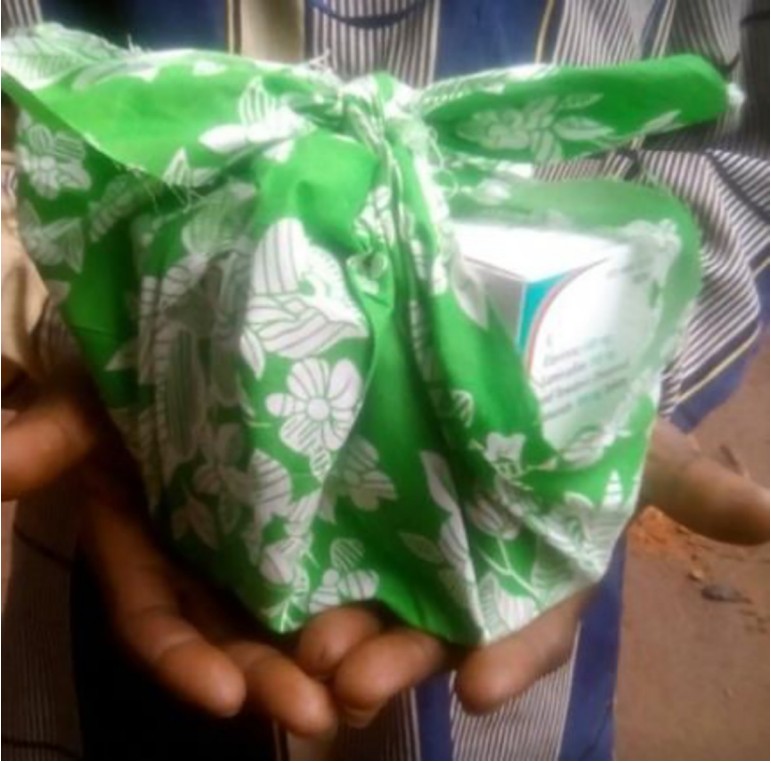

**Fig 3. Run-away bags including 3 months of ARV's for patients under contingency plan.**

## Acceptability

Acceptability of the strategy was measured after interviewing members of the general community, and direct beneficiaries of the program. Semi-structured interviews assessed the timing (time of day, day of the week) of the services, their level of comfort with physical set-up, and comfort with community health workers who were providing the services. The questions were answered on 6-point scale (strongly agree, agree, slightly agree, slightly disagree, disagree, strongly disagree) followed by a room for comments. Interviews were conducted by individuals who were not part of the program. The questions were translated into the local language spoken in the 5 locations (Payams) where interviews were conducted. Interviews were conducted in private houses. The aspect of the program was determined acceptable when interviewees strongly agreed or disagreed with the test & start strategy.

## Results

From September 2015 to June 2018, 14824 people were counselled and accepted to be tested and 498 (3.3%) of them tested HIV positive. While 130 (40.4%) had been tested before in other clinics, 8 (2.4%) patients were already on ART at the time of HCT.

Among the 498 confirmed HIV positive patients, 395 (79.3%) accepted to start ART; 260 (65.8%) were women and 10 (2.5%) were children under 15 years old. Median age at ART initiation was 33 years old [IQR 26–41]. Among the 103 patients who didn't start on ART only 54 of them could be interviewed before the end of the study, the main reasons reported to not start ART were: not ready / need of consulting with a relative / preference to start ART at hospital level / already on ART. Baseline CD4 count was available for 372 (94%) of the patients

**Table 1. Patients' demographic and clinical characteristics at enrolment in test and start program.**

| Characteristics | <15, n = 10 | 15–29 years, n = 128 | 30 years and more, n = 257 | Total, n = 395 |
|---|---|---|---|---|
| Age, (Median age, [IQR]) | 2, [1–7] | 24, [20–27] | 38, [33–45] | 33, [26–41] |
| Female, n (%) | 5 (50) | 96 (75.0) | 159 (61.8) | 260 (65.8) |
| Marital status | na | | | |
| married | | 59 (46.5) | 124 (48.2) | 183 (47.7) |
| not married | | 68 (53.5 | 133 (51.8) | 201 (52.3) |
| missing | | 1 | | 1 |
| Prior HIV tested | | | | |
| n (%) | 2 (25.0) | 41(39.4) | 87(41.4) | 130(40.4) |
| missing | 2 | 24 | 47 | 73 |
| On ARV before | | | | |
| n (%) | 2(25.0) | 1(0.9) | 5(2.4) | 8(2.4) |
| missing, n | 2 | 20 | 44 | 66 |
| Clinical WHO stage, n (%) | | | | |
| 1 | 6(60.0) | 94(73.44) | 141(55.1) | 241(61.0) |
| 2 | 1(10.0) | 24(18.7) | 77(30.1) | 102(25.8) |
| 3 | 2(20.0) | 8(6.3) | 31(12.1) | 41(10.4) |
| 4 | 1(10.0) | 2(1.6) | 7(2.7) | 10(2.5) |
| Missing, n | | | 1 | 1 |
| Tuberculosis diagnosis, n (%) | | | | |
| No sign | 6(100.0) | 107(100.0) | 214(96.8) | 327(97.9) |
| TB suspects | | | 7(3.2) | 7(2.1) |
| *Tested positive* | | | *1* | *1* |
| *Already on TB treatment* | | | *1* | *1* |
| Missing, n | 4 | 21 | 36 | 61 |

with a median CD4 count of 425 cells/μl [IQR 262–620]; 172 (46.2%) and 57 (15%) of the patients had a CD4 count below 500 cells/μl and 200 cells/μl, respectively. WHO clinical stage was assessed for all patients, 51 (13%) patients had WHO stage 3–4. All patients were clinically screened for TB and 7 (2.1%) were referred to YSH for further TB diagnostic and treatment. At the first consultation, 113 patients (38.9%) had BMI < 18.5. Patient's clinical characteristics at enrolment are summarized in Table 1.

Among the patients starting ART, 336 (85.7%) were initiated on the same day while 44 (11.2%) were initiated within the following 7 days after diagnosis, 5 (1.3%) were initiated in the second week after diagnosis and 7 patients (1.7%) started later than two weeks. Three hundred and sixty five (92.4%) patients were started on WHO recommended first line regimen (TDF/3TC/EFV), 28 (7.1%) patients were started on AZT/3TC/NVP due to low creatinine clearance. While 8 patients under 15 years old were started on AZT/3TC/NVP, 2 other children in same age group started on ABC/3TC/NVP as LPV/r was not available in the country. Adherence was routinely recorded in all consultations and only 72 (18%) self-reported being non-adherent.

Adverse events were checked in all consultations and a total number of 138 adverse events were reported in 85 patients. The most frequent side effects were dizziness 35 (25%) and headache 17 (12.3%). Two patients on TDF-based regimen developed low creatinine clearance and were switched to ABC-based regimen. Twenty-one patients were referred to YSH for further investigations being TB suspicion the main reason in 6 patients (28.6%).

## Patients' outcomes at 12 and 18 months of follow up

At 12 months of follow up 236 (67.5%) patients were still under care in the program; 56 (16%) patients were lost to follow up, 9 (2.6%) were dead and 49 (14%) were transferred out or handed over. At 18 months, 262 (66.3%) patients were still under care, 116 (29.4%) patients were lost to follow up (LFU) and 17 (3.4%) died (Table 2). Kaplan-Meier estimated RIC at 12 and 18 months was 80.6% [95% CI: 75.9–84.5%] and 69.9% [95% CI: 64.4–74.8%] respectively (Fig 4). No differences in RIC were observed by baseline CD4 count. A multivariate complete case cox regression estimated that not being married hazard ratio, HR 2.5, $p = 0.001$) and baseline WHO stages 3–4 were associated with attrition (HR 2.2, $p = 0.014$).

Viral suppression at baseline and after 12 months and end of program months was defined as an HIV RNA measurement of less than 1000 copies/mL following WHO guidelines. Viral load at 12 months was available for 197 (56%) of patients initiated on ART and overall data on viral suppression at the end of the study was available for 272 (78%) of the patients. At 12 months, 178 (90.3%) of the patients with available viral load had attained viral suppression (Table 3). Factor associated with detectable viral load at 12 months was being single (OR 2.5 [95% CI: 1.3–5.1], ($p = 0.008$); when data was disaggregated by baseline CD4 count, no difference was observed.

The programme HIV care cascade shows 79.3% of ART initiation, 80% RIC and 91.2% of viral suppression (Fig 5).

## ART outcomes in CAGs patients

Seventy two patients on ART were included in 26 different CAGs, which represent a mean of 3.7 patients per CAG. Data on outcomes was available for 72 (73.4%) patients at the end of the study. At 12 months of follow up, 4 (5.6%) patients were LTFU, 3 (4.2%) were handed over to another program, no deaths were recorded and 65 (90.3%) were still under follow up. At 18 months 66 (91%) patients were handed over to MoH, 6 (8.3%) patients were LTFU and no deaths were recorded. RIC for patients in CAGs at 12 and 18 months were 94.3 [95% CI: 85.2–97.8] and 90.9 [95% CI: 80.8–95.8], respectively. RIC in patients on CAG was significantly higher (90.9% vs 63.4%, p <0.001) than patients under regular follow up by the mobile teams (non-CAG). (Fig 6).

## Outcomes for patients following contingency plan

During the 3-year project, contingency plan was activated 9 times due to active conflict in the area and provided to 101 patients in 6 different locations with ARV refill for 3 months distributed as "run-away bags". By the end of the study, 18 (17.7%) of these patients were LTFU and 2 (1.9%) reported dead. Kaplan-Meier estimated that overall RIC of these patients was 87.8% [95% CI: 79.5–92.9] which was significantly higher than for the regular cohort followed up by

**Table 2. Patients' outcomes at 12 and 18 months.**

| Outcomes at 12 and 18 months of follow up, N = 350 | | |
|---|---|---|
| **Outcomes** | **At 12 months** | **At 18 months** |
| **All patients, (n = 350)** | | |
| Handover | 26 (7.4) | 180 (51.4) |
| Lost to follow-up, n (%) | 56 (16.0) | 80 (22.9) |
| Transferred out, n (%) | 23 (6.6) | 77 (22.0) |
| Dead, n (%) | 9 (2.6) | 13 (3.7) |

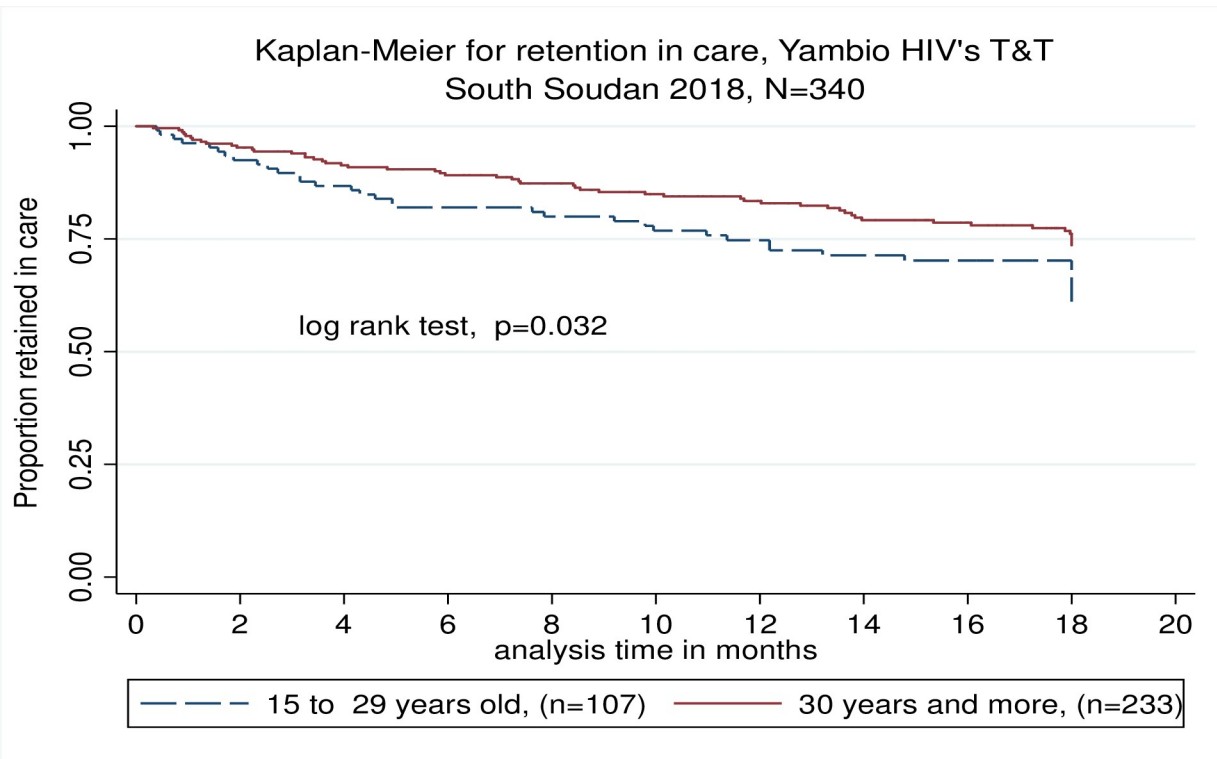

**Fig 4. Kaplan Meier estimated RIC at 12 and 18 months.**

the mobile teams ($p = 0.014$) (Fig 7). Viral suppression for these patients at 12 and 18 months was 90% and 82%, respectively.

## Acceptability of the program

In order to assess the community perception and acceptance of the test and start program it was planned to interview 341 patients and community members from all the five locations, and team has achieved to do 279 (82%) interviews. One site (Li-Rangu) was not accessible for the interview team due to ongoing conflict at the time of the assessment.

A questionnaire to evaluate acceptability of the programs was done in 279 patients in 5 different *payams (counties)*. Among them 274 (98.2%) had heard about the test and start program mainly through MSF informative sessions (53.2%), mouth to mouth (19.6%) and radio (16.9%). Most of the interviewed (69.3%) had good knowledge of the test and start program and the majority (84.4%) believed that the test and start services were beneficial for the community.

**Table 3. Viral load suppression at 12 months and overall.**

|  | At 12 months | Overall |
|---|---|---|
| *Based on available viral load* | | |
| **VL available** | 197 | 272 |
| **Viral suppression** | 178 | 248 |
| **%** | 90.3 | 91.2 |
| **95% CI** | [85.8–94.2] | [87.9–94.5] |

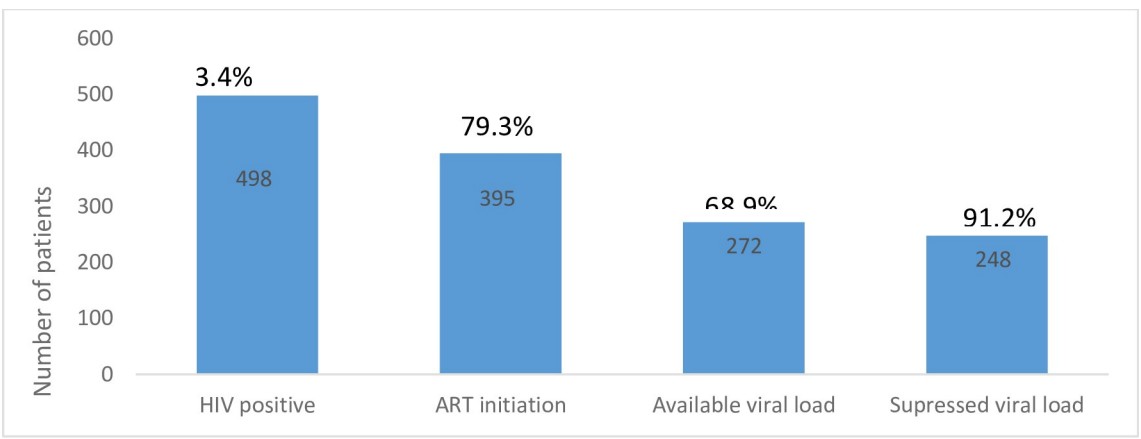

**Fig 5. HIV care cascade at Yambio Test and Start program, January 2015 –June 2018.**

When asked about the quality of the services, 95.3% of the patients strongly agreed that the services were provided at the right time of the day; 84.3% of patients were comfortable or very comfortable with the physical emplacement and set-up and 81.6% strongly agreed or agreed that it was good to have services outside the health centre.

When asked about disadvantages of the program for the community, vast majority (96.1%) reported no disadvantages. We asked 247 patients, to whom test and start services were proposed, about their level of satisfaction regarding a) having CHWs involved in the program, b) the work done by the counsellors and c) their agreement to get tested for HIV. The majority of

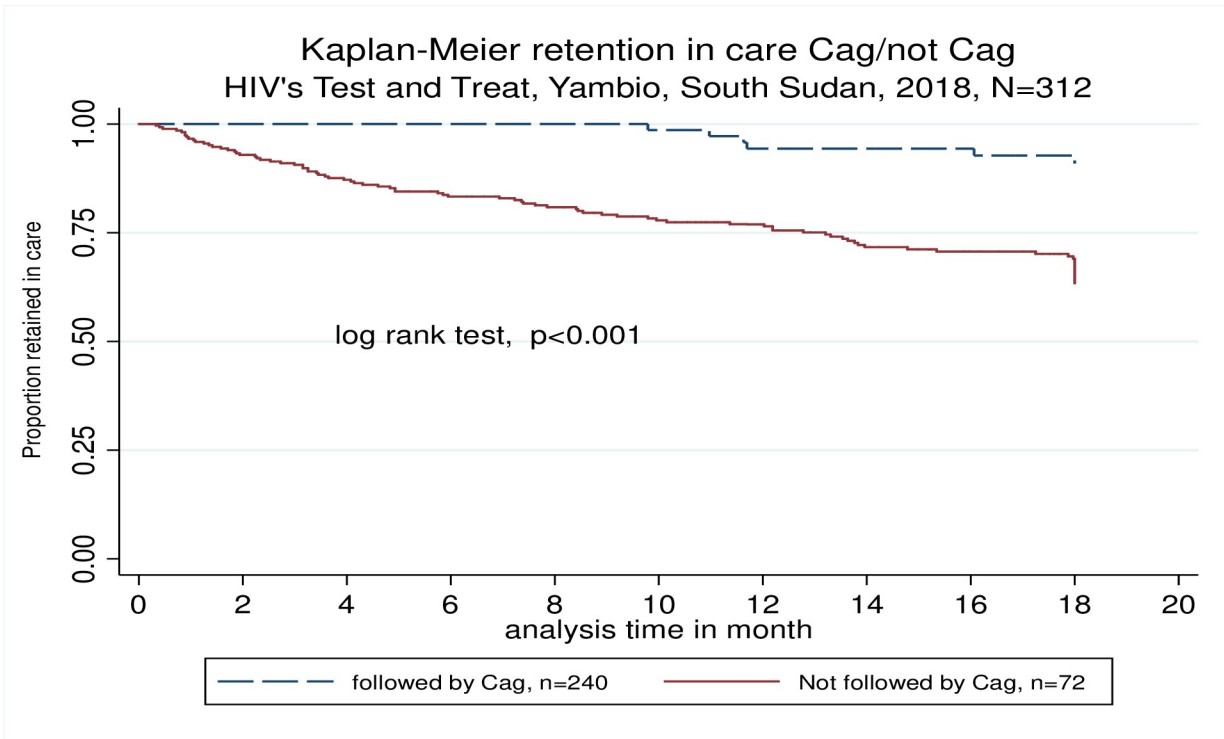

**Fig 6. Kaplan-Meier RIC for patients under CAGs compared to patients under regular follow up.**

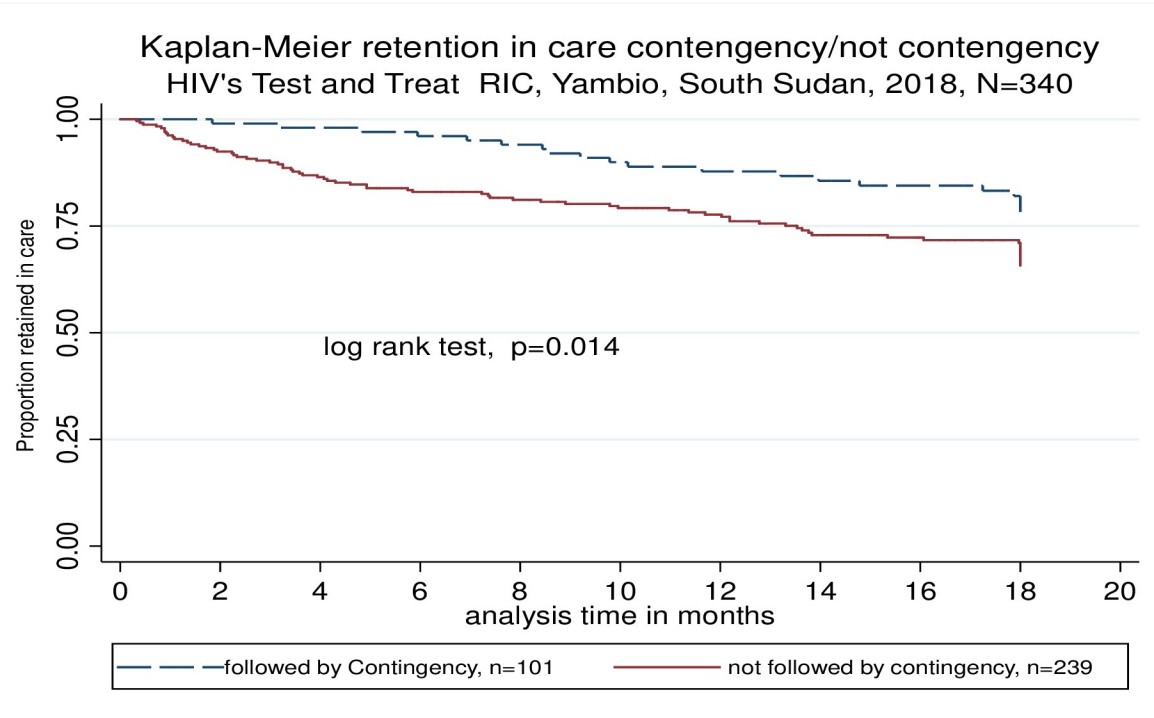

**Fig 7. Retention in care for patients on contingency plan compared to patients under regular follow up.**

the respondents, 229 (92.7%) agreed or strongly agreed that having a CHW was good for them and that helped them having treatment in confidence and safety; 240 (97.2%) agreed or strongly agreed that counsellors did a good job explaining the services provided, including lab results; and 243 (98.4%) finally agreed to get tested for HIV.

When requested to provide advice to MSF to improve test and start services, interviewed people mentioned that "We need testing to be continuous without stopping", or "what I want from MSF is let them remain and help us with our drugs", or "we need the handover partner to follow MSF footsteps" or "MSF should increase the year of the project for test and treat because most of the people are still not in access to the services".

## Discussion

Our study shows RIC and viral suppression results similar to those observed in stable contexts and with HIV services provided at a health care facility [8, 28, 29].

It has been reported that community HIV testing and counselling had high coverage and uptake and identified HIV-positive people at higher CD4 counts than facility testing. This may be also related with the reasonably good immunological and clinical situation compared to other settings probably due to the active HCT campaign, the remoteness of the region and the conflict situation in the area [30]. CAGs and CHCWs were key in supporting patients on ART as shown by the retention in care and viral suppression in this group. CAGS have been effective in other HIV programs at rural areas but in our study shows that it can also be effective in unstable contexts [31, 32].

Majority of our patients (67.5%) accepted ART initiation on the same day of diagnostics and reached close to 80% over the subsequent few weeks. However still 20% of patients refused to start ART being the main reasons "the need to discuss with a family member "or "not feeling

ready". In our program linkage to care was not a barrier as the teams were already in the community and there was no need to refer patients for ART initiation to a health facility. Acceptability of HIV services was high owing to having HIV services close to homes in the community. Before the program was implemented there were concerns about potential low uptake or poor adherence among healthy patients with high CD4 counts, especially in resource-limited settings [33]. However, our program showed that despite high level of median CD4 count, ART uptake was high in this population.

Patient's outcomes including virological suppression and retention in care in our program were comparable to those in other facility-based HIV programs in the region without the security constrains of South Sudan [16, 23, 24, 34, 35]. Similarly high rates of RIC and virological suppression were observed in both patients followed up by the mobile teams and those who underwent contingency plan, which suggests that community based HIV services are feasible and suitable for conflict affected population with reasonably good program outcomes.

Some of the limitations of our studies are related with the challenge to get strong data recording in spite of the context. The acceptability assessment could not be more complete due to the challenge to trace back most of the participants.

Some of the strengths of the study are the uniqueness of the setting and the availability to implement CAGs and contingency plans in periods of instability. Including CHWs as part of the strategy shows a sustainable manner to reproduce this project in other similar settings.

## Conclusions

Community based HIV services could help to increase ART coverage and retention in care in settings where access to these services is still poor or disrupted by humanitarian crisis. HIV services can be included since the beginning in any humanitarian response if community-based HCT and ART initiation is used. CAGs and the inclusion of CHWs were an important component of our study due to the difficulties from patients to reach the mobile clinics. Our study shows that provision of HIV services at community level can be done with good programme outcomes suggesting that it could be replicated in similar settings to ensure PLWHIs are included in the health response in conflict affected settings.

## Author Contributions

**Conceptualization:** Cecilia Ferreyra.

**Data curation:** Fara Wagbo Temessadouno.

**Formal analysis:** Fara Wagbo Temessadouno.

**Investigation:** Cecilia Ferreyra, Beatriz Alonso, Vicente Descalzo-Jorro.

**Methodology:** Cecilia Ferreyra, Buai Tut, Vicente Descalzo-Jorro.

**Project administration:** Cecilia Ferreyra, Laura Moretó-Planas, Beatriz Alonso, Vicente Descalzo-Jorro.

**Supervision:** Cecilia Ferreyra, Laura Moretó-Planas, Beatriz Alonso, Buai Tut, Victoria Achut, Mohamed Eltom, Endashaw M. Aderie.

**Validation:** Cecilia Ferreyra, Laura Moretó-Planas, Victoria Achut, Mohamed Eltom.

**Visualization:** Cecilia Ferreyra.

**Writing – original draft:** Cecilia Ferreyra.

**Writing – review & editing:** Cecilia Ferreyra, Laura Moretó-Planas, Fara Wagbo Temessa-douno, Beatriz Alonso, Buai Tut, Endashaw M. Aderie, Vicente Descalzo-Jorro.

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
