## [Decision Letter · Decision Letter 0]

17 Apr 2020

PONE-D-20-06492

Evaluation of a community-based HIV test and start program in a conflict affected rural area of Yambio County, South Sudan

PLOS ONE

Dear Dr Ferreyra,

Thank you for submitting your manuscript to PLOS ONE. After careful consideration, we feel that it has merit but does not fully meet PLOS ONE’s publication criteria as it currently stands. Therefore, we invite you to submit a revised version of the manuscript that addresses the points raised during the review process.

We would appreciate receiving your revised manuscript by Jun 01 2020 11:59PM. To enhance the reproducibility of your results, we recommend that if applicable you deposit your laboratory protocols in protocols.io, where a protocol can be assigned its own identifier (DOI) such that it can be cited independently in the future. For instructions see: http://journals.plos.org/plosone/s/submission-guidelines#loc-laboratory-protocols

We look forward to receiving your revised manuscript.

Kind regards,

Joel Msafiri Francis, MD, MS, PhD

Academic Editor

PLOS ONE

Journal Requirements:

1. Thank you for including your competing interests statement; "The authors have declared that no competing interests exist."

We note that one or more of the authors are employed by a commercial company:

Épicentre

Reviewers' comments:

Reviewer's Responses to Questions

**Comments to the Author**

1. Is the manuscript technically sound, and do the data support the conclusions?

Reviewer #1: Yes

Reviewer #2: Partly

2. Has the statistical analysis been performed appropriately and rigorously? 

Reviewer #1: Yes

Reviewer #2: No

3. Have the authors made all data underlying the findings in their manuscript fully available?

Reviewer #1: Yes

Reviewer #2: No

4. Is the manuscript presented in an intelligible fashion and written in standard English?

Reviewer #1: Yes

Reviewer #2: No

5. Review Comments to the Author

Reviewer #1: Thank you for putting together an interesting manuscript on program data from Sudan. Most of the outcomes - retention in care, ART initiation in the test and treat era have been described in other papers. The uniqueness of the setting in Sudan makes this manuscript important. The manuscript is well written, and highlights some innovation like the contingency plan bags. Please some comments below to improve clarity on some of the issues.

Abstract

Background - last line, remove ad after the and

Results: CD4 count is no longer a key factor in HIV management in the ear of test and treat, please remove that and focus on factors associated with attrition in care.

Main Manuscript

Lines 78-79 - Rephrase sentence, HIV testing leads to access to ART for infected people and prevention services for the uninfected.

Lines 98-99 - Include a reference

Context and Rationale

Lines 126, 127 remove decimal separator (point) in 3000 and 1500. In general you need to be consistent in how you write the numbers throughout the manuscript including the abstract. In some instances there is a comma, other a point/full stop and in others there is nothing. Decide on one method, preferably without separators as that is what is usually used in general numbers, and decimal separators applied in accounting.

Methods

General: Include the acceptability assessment in the method section. That would also shorten the results section on acceptability

Lines 174 - correct the tense of the sentence

Line 240 - 21 patients referred for TB treatment, is it new TB diagnosis after ART initiation or is the same one referred to in Table 2 - that are already on TB treatment?

Line 286 - consistency in the name - T&T services

Discussion

Lines 306 - 310: Sentence is not clear in terms of what were the findings in your study and how does it compare with literature.

Please also include the role of CAGs and CHCWs in supporting patients to stay in care in the discussion as I think that is one of the major findings of the study. You can do some reference to the importance of assisted self-management principles for chronic disease management.

Reviewer #2: 1. The paper would benefit from grammatical and style editing.

Introduction

2. Background on the HIV context and HIV service provision in conflicts settings should be presented earlier in the section. Specifically, I would rearrange the section as follows: (i) epidemiological context in Southern Sudan and other conflict settings (Paragraph 6), (ii) availability of HIV testing and ART services (Paragraph 4-5), (iii) evidence on community-based and test-and-start interventions (Paragraph 2-3).

3. Lines 69-77: I would suggest removing or significantly shortening the first paragraph. The language is also very inflated. For example, “Entire regions are falling far behind and vulnerable population in all countries— whether they are high-income, middle-income or low- income—a common pattern has emerged: gains on HIV, health and development have overlooked the people in greatest need” could be revised to “Coverage in access to HIV testing and ART services has remained disproportionately low in vulnerable and marginalized populations.”

4. Lines 79-81: Where is the evidence from and are they from settings similar to Southern Sudan? Providing such detail would help to contextualize what evidence is available and how the current study addresses existing research gaps.

5. Lines 109-11: The program involves more than just community-based testing and universal ART, but also community groups for ART patients. The study should therefore be characterized as a community-based testing, treatment and adherence program.

6. Lines 120-129: What is the size of the catchment population serviced by the program?

Methods

7. The section provides inadequate description of data collection and analysis procedures. For example, sampling and data collection for the acceptability survey are not described. Data collected at each stage of the program should be clarified. A section on data analysis is also missing. Further, description of the intervention could be much more concise to allow for better explanation of data collection and analysis procedures.

Results

8. Table 1: It is not clear why data are stratified by age group. If the author is interested in understanding differences in outcome by age groups, this should be clearly stated as a research objective.

9. Lines 218-219: What were the reasons why HIV-positive clients did not start on ART?

10. Tables 1 and 2 should be combined to present both sociodemographic and clinical characteristics.

11. Lines 243-250: The results of the regression analysis for attrition at 12 and 18-months should be presented in a table. Further, the results should be described consistently. For example, lines 243-245 pertain to patient retention at 18 months. Lines 245-247 pertain to patient retention at 12 and 18 months. Lines 248-250 conversely pertains to patient attrition. It is also not clear whether the outcome relates to patient outcomes as 12 or 18 months.

12. Lines 255-259: Similarly, the results of the regression analysis for viral suppression should be detailed in a table and described consistently.

13. Lines 260-265: The sample for pregnant women seems too small for meaningful statistical analysis.

14. Lines 266-280: The sub-sections should be merged to present the results of clinical outcomes by follow-up intervention. Further, perhaps the survival curves (Figures 5 and 6) should be combined.

15. Figure 4: The figure presents interesting data but is not referenced in the text.

Discussion

16. The section requires substantial revision. First, discussion on patient characteristics associated with adherence and viral suppression should be included. Second, findings should be interpreted with caution. For example, “very few refused to start on ART” is purported despite 20% of HIV-positive patients not starting on ART. Third, reasons why patients did not start on ART or were not retained in care should be discussed. Fourth, more substantial reflection on the strengths and limitations of the study should be added.

6. PLOS authors have the option to publish the peer review history of their article (what does this mean?). If published, this will include your full peer review and any attached files.

Reviewer #1: Yes: Limakatso Lebina

Reviewer #2: No

---

## [Author Response · Author response to Decision Letter 0]

7 Sep 2020

Dear Editors

PONE-D-20-06492R1

Evaluation of a community-based HIV test and start program in a conflict affected rural area of Yambio County, South Sudan

Dear Zsoka Murakozi

PLOS ONE 

Thank for your feedback. 

We have revised all the comments which have been addressed in the manuscript. 

Kind regard,

Dr Cecilia Ferreyra

---

## [Decision Letter · Decision Letter 1]

20 Oct 2020

PONE-D-20-06492R1

Evaluation of a community-based HIV test and start program in a conflict affected rural area of Yambio County, South Sudan

PLOS ONE

Dear Dr. Ferreyra,

Thank you for submitting your manuscript to PLOS ONE. After careful consideration, we feel that it has merit but does not fully meet PLOS ONE’s publication criteria as it currently stands. Therefore, we invite you to submit a revised version of the manuscript that addresses the points raised during the review process.

We look forward to receiving your revised manuscript.

Kind regards,

Joel Msafiri Francis, MD, MS, PhD

Academic Editor

PLOS ONE

Reviewers' comments:

Reviewer's Responses to Questions

**Comments to the Author**

1. If the authors have adequately addressed your comments raised in a previous round of review and you feel that this manuscript is now acceptable for publication, you may indicate that here to bypass the “Comments to the Author” section, enter your conflict of interest statement in the “Confidential to Editor” section, and submit your "Accept" recommendation.

Reviewer #1: All comments have been addressed

Reviewer #3: (No Response)

2. Is the manuscript technically sound, and do the data support the conclusions?

Reviewer #1: Yes

Reviewer #3: Partly

3. Has the statistical analysis been performed appropriately and rigorously? 

Reviewer #1: Yes

Reviewer #3: No

4. Have the authors made all data underlying the findings in their manuscript fully available?

Reviewer #1: Yes

Reviewer #3: No

5. Is the manuscript presented in an intelligible fashion and written in standard English?

Reviewer #1: Yes

Reviewer #3: Yes

6. Review Comments to the Author

Reviewer #1: The manuscript flow has improved and some edits are still required.

Introduction

Lines 72-73 - which country, please provide reference.

Lines 74 - reference

Lines 78 - reference

Line 78 - The section on the MSF program activities - should be a new paragraph

Line 78 - First use of MSF - provide full name before you start using the abbreviation

Methods

Line 149 - Incomplete sentence : follow-up and f??

Results

Lines 261 - 263 - retention is said to be higher at 12 months, but more people in care at 18 months (236 at 12 months and 262 at 18 months); and both timepoints its out of 350 (Table 2).

Table 3 - Not necessary - all information explained in the text.

Lines 275-277 - Should be explained in the methods section

Lines 309 - 310 - should be explained in the methods section

Discussion

Lines 344 - 346: Compare the update of ART to other programs, as well as the reasons.

One publication on why people living with HIV do not initiate ART

Katz, I. T., Dietrich, J., Tshabalala, G., Essien, T., Rough, K., Wright, A. A., Bangsberg, D. R., Gray, G. E., & Ware, N. C. (2015). Understanding treatment refusal among adults presenting for HIV-testing in Soweto, South Africa: a qualitative study. AIDS and behavior, 19(4), 704–714. https://doi.org/10.1007/s10461-014-0920-y

Conclusion

Line 369 Please revise and remove from the beginning or consider: "HIV services should be included as part of any humanitarian response."

Reviewer #3: 1. This is a first time I have reviewed this manuscript, so I could not provide an answer to whether my previous comments were adequately addressed.

2. Some statistical methods are not described for some of the presented results.

3. Data access was restricted but an email to request for the data has been provided

7. PLOS authors have the option to publish the peer review history of their article (what does this mean?). If published, this will include your full peer review and any attached files.

Reviewer #1: No

Reviewer #3: No

---

## [Author Response · Author response to Decision Letter 1]

19 May 2021

Summary: The study aims at improving HIV testing and ART coverage among people living in a conflict affected community of South Sudan in Africa. The investigators offered HIV counselling and testing in the study area and started on ART whoever tested HIV positive and was willing to do so. This is particularly an important study especially that it was conducted in a conflict affected community where access to HIV care may be challenging. The work in this manuscript is worth publishing and it fits very well within the UNAIDS strategy of 90-90-90 target of year 2020. However, I have some major and minor comments that need to be addressed before this work can be published.

Abstract: 

Line 34, abbreviation for CAGs is presented for the first time before a full text is given. 

CF: the term is now written in full and abbreviation put under bracket,

Line 36, you mention “RIC was significantly higher at 18 months in patient under contingency plan (90% p<0.001) when compared to patients on regular follow up”. First, “patient” should be plural, i.e, “patients”. Also, you present one statistic in the comparison without giving the other one, which I find hanging. I suggest this be revised as follows “RIC was significantly higher at 18 months in patients under contingency plan when compared to patients on regular follow up (90% vs xx%, p<0.001)” where xx is RIC for patients on regular follow up. 

CF: the word “patient has been changed to “patients” in line 36. Regarding your comment on RIC, the sentence has been modified as follow in line 40: RIC was significantly higher at 18 months in patients under CAGs when compared to patients on regular follow up (90.9% vs 63.4%, p,0.001)”

In line 35-36, the 95% confidence interval is not appropriately presented, I would reorder to read as xx% [95% CI: xx-xx%]. This is the same issue even in the results section.

CF: the 95% CI is now presented as suggested in both the abstract and the results sections.

Under conclusion in line 40, the authors mention “Our study shows a high level of acceptance to HCT… “. This conclusion is not supported by the findings. It remains unclear how acceptance was measured, moreover, no summary finding about acceptance is presented in the abstract.

CF: thanks for the comment, we have added a line in the abstract about the acceptability part and then explained further in methodology and results, also the sentence about acceptance has been modified as follows “Our study shows that HCT and early ART initiation in conflict affected populations can be provided with good program outocomes”..

Introduction:

Lines 72-73, please provide references for the following statements, “… and ART coverage in the country is around 10%” and “An estimated 16000 new HIV infections happen every year and about…” 

CF: the ART coverage and number of new infections have been updated and the following references have been included:

- Jervase A et all, UNAIDS report 2018 and UNHCR South Sudan – Conflict, displacement, famine and the HIV response, Time to act! 

- UNAIDS. HIV/AIDS, South Sudan, 2019 [Internet]. 

In line 103, a terminology “HIV testing and counselling (HTC)” is used yet under abstract in line 28 a statement “HIV counselling and testing (HCT)” is used. I suggest that for consistence, one terminology be used throughout.

CF: we have now kept it consistent as HIV counseling and testing (HCT) throughout the manuscript.

Methodology:

In line 135, “All patient data was …”, data is plural. So, replace “was” with “were”. This needs to be observed throughout the manuscript. Also, in line 136, replace “where” with “were”. Note that Epidata is meant for data management not analysis. I would revise the statement “… paper forms were later on entered into Epidata database for further data management” (lines 136-137).

CF: “all patients data” sentence has been changed to plural and revised throughout the manuscript. The Epidata usage is now appropriately mentioned as for data management.

In line 141, you say the main outcome of interest was acceptability. I suppose you meant acceptability to HCT/HTC (please clarify). This being a main outcome, a clear methodology on how acceptability was measured and analysed should be presented. The focus seems to be on retention (as if it’s the main outcome) rather than acceptability.

CF: the aim of the study was to assess feasibility of community-based HIV care in this conflict affected population, which was measured through programme outcomes (RIC, LFU, viral suppression and mortality). A qualitative component of the programme included programme acceptability interviews in the community and direct beneficiaries of the programme. We have modified this paragraph as follow: “the aim of the study was to describe programme outcomes which included retention in care, viral suppression, LFU and death at 12 and 18 months of programme initiation.”

As per the acceptability part, we modified the paragraph in line 148 as follow: “A semi-qualitative assessment using questionnaires was done at community level to assess acceptability, including participants as follows:…”

In line 144, you mention “… factors associated with attrition”. While it’s true that attrition is a complement of retention, I would focus on analysing retention not attrition. 

CF: the sentence has been modified to: “…factors associated with RIC.”

Results:

In line 237, a closing bracket in [IQR 262-620 is missing

CF: corrected, bracket closed.

In Table 1, a comma was used for % values instead of a decimal point. For example, under female category, the percentage is presented as 75,0 instead of 75.0. This is the same issue throughout Table 1.

CF: all commas in Table 1 where change to decimals.

In line 262, “loss” should be “lost”

CF: now corrected as LFU

In line 260, I would replace “Patient’s” with “Patients’ ”

CF: this is now corrected to “Patients’”

In line 264, you present OR without first giving what this stands for. I suppose this stands for odds ratio. Also, no methodology presented on how mortality was analysed, for example, what statistical method was used to obtain the OR for mortality. Also, present the 95% CI for the OR throughout the results section.

CF: the mortality analysis was not found to be relevant for the discussion, hence the sentence has been deleted to kept consistency in reporting other programme outcomes (RIC, viral suppression and LFU) as suggested by the reviewers. 

In line 286, you say “a multivariate complete case Cox regression estimated that not being married (OR 2.5, p0.001) …” Here it’s strange how you present OR from a Cox regression. Note that, Cox regression provides hazard ratio (HR) not OR. Please check your analysis methods. Also, please briefly describe how survival time was calculated, was any censoring done?

CF: thanks for highlighting this error. Indeed, we calculated Hazard Ratios (HA) and not Odds Ratios (OR). The analysis method has been clarified and the sentence has been modified in line 274. 

In line 268, “p 0.001” should be “p=0.001”. This is an issue almost everywhere in the manuscript. Please add the equal sign between p and the value throughout.

CF: this is now corrected throughout to mean “p=0.0x”

In line 279, the sentence “… were and 18 …” is not clear. I think some word is missing between “were” and “and”

CF: this is now corrected; it was a mistake from the previous version.

In line 280, you present analysis on factors associated with detectable viral load, however, the methodology for analysing these factors is not described/mentioned in the methods section. 

CF: a sentence to clarify the method to analyse viral load has been included in methods in line 149 as follow: “We used a stepwise backward regression technique at p-value<0.2 to include variables in risk factor assessment model. All variables with p-value >0,2 were sequentially removed in the process; a p-value <0.05 was used for inclusion of factors in the final model”

In Table 3, in the last row, “CI95%”, should be “95% CI”

CF: corrected.

In lines 291-292, you say “At 12 months, 4 were LTFU, no death, 65 still in follow up”. This adds up to 69. Were all 72 available at 12 months? If yes, what happened to the 3 patients?

CF: the 3 patients were handed over to other program during the period, which is now included in the text in line 298.

In line 294, you say “RIC was significantly higher (90% at 18 months, p<0.001) when compared to patients on regular follow up…”. It’s not clear what group of patients the statement refers to. Also, present RIC for patients on regular follow up and the test statistic used to compare the two.

CF: we are comparing RIC in CAGs and non-CAGs/”patients under regular follow up” with the mobile teams and included the statistics to compare both. Kaplan Meir was used to measure RIC. The sentence has been modified as follows: “RIC in patients on CAG was significantly higher (90.9% vs 63.4%, p <0.001) than patients under regular follow up by the mobile teams (non-GAG) “.

In Fig 6 (as well as Fig 4 and Fig 7), I suggest you revise the y-axis label to "Proportion retained in care" drop % since the y-axis is showing the proportion not percentage. Also, in Fig 6, use p<0.001 (not p-value<0.001) for consistence, x-axis should indicate unit of measurement for analysis time (I think months), let one line be dotted or long-dashed for readable print out (black and white), please add number at risk (also for Fig 4 and Fig 7).

CF: This is now corrected, and improved as per the suggestions in Fig 4, 6 and 7..

In line 304, I think “p 0.014” is showing the overall difference in retention but not the difference at 12 or 18 months. Please clarify and revise accordingly. Also, it's not clear what test statistic was used for this comparison.

CF: thanks for the remark, indeed the figure is showing overall RIC and not the difference, this has now been clarified in the manuscript as follow: “Kaplan-Meier estimated that overall RIC of these patients was 87.8% [95% CI: 79.5-92.9] ,which was significantly higher than for the regular cohort followed up by the mobile teams (p=0.014). The statistic test used for the comparison was logrank test, and this has been updated in the text accordingly.

Lines 308-330, if acceptability is the main outcome as mentioned in the methodology, then I would present this before retention.

CF: as clarified before the main outcomes of the study were those related with RIC. We have included a paragraph in the methodology section, line 235, to better explain the acceptability component. 

Line 309, it’s not clear how the number 279 was determined

CF: a sample size of 341 participants was initiatly estimated, due to the security constrains for the area the team managed to perform 279 (82%) of the interviews. This has been now included in line 334-336. 

Line 315, “a” should be “at”

CF: thanks for the remark, it has now been changed. 

Fig 5 is faint and of low quality. The label on third bar is cut off. Since the figure shows both numbers and percentages, I suggest a two-y-axis figure, one axis showing numbers and the other one showing the percentage.

CF: figures have been revised as suggested above.

Discussion:

Line 332, you say “our study shows a high level of acceptance to HCT…”, I am struggling to see how this is shown by the study findings. What result are you basing on to claim this?

CF: as suggested above we have modified the the main outcomes as follows: “our study shows RIC and VL suppression results similar to those observed in stable contexts and with HIV services provided at a health care facility”. 

Line 355-365, you say “Patients characteristics associated with better adherence …”, I did not see any analysis and results on adherence before this statement. Also, it's not mentioned how adherence was measured. Please clarify.

CF: thanks for the comment. We have remove the sentence to avoid confusions. 

Conclusion:

Lines 367-372, again, what was the main outcome of this study? In the methodology, line 141, the main outcome is acceptability, but no conclusion is presented on this.

CF: as mentioned above the main outcome was RIC and viral suppression, appart from death and LFU. This has been modified along the manuscript. 

Lines 371-372, you say “Our study shows that differentiated models of care are feasible in these contexts”. I find this conclusion inappropriate since it is not supported by your findings. The study did not study the feasibility of differentiated models but rather community HCT and ART initiation.

CF: Thanks for your comment, we have modified the sentence to clearly indicate the main purpose of the study as follows: “Our study shows that provision of HIV services at community level can be done with good programme outcomes suggesting that it could be replicated in similar settings to ensure PLWHIs are included in the health response in conflict affected settings”.

---

## [Decision Letter · Decision Letter 2]

25 Jun 2021

Evaluation of a community-based HIV test and start program in a conflict affected rural area of Yambio County, South Sudan

PONE-D-20-06492R2

Dear Dr. Ferreyra,

We’re pleased to inform you that your manuscript has been judged scientifically suitable for publication and will be formally accepted for publication once it meets all outstanding technical requirements.

Kind regards,

Joel Msafiri Francis

Academic Editor

PLOS ONE

Additional Editor Comments (optional):

Reviewers' comments:

Reviewer's Responses to Questions

**Comments to the Author**

1. If the authors have adequately addressed your comments raised in a previous round of review and you feel that this manuscript is now acceptable for publication, you may indicate that here to bypass the “Comments to the Author” section, enter your conflict of interest statement in the “Confidential to Editor” section, and submit your "Accept" recommendation.

Reviewer #3: All comments have been addressed

Reviewer #4: All comments have been addressed

2. Is the manuscript technically sound, and do the data support the conclusions?

Reviewer #3: Yes

Reviewer #4: Yes

3. Has the statistical analysis been performed appropriately and rigorously? 

Reviewer #3: Yes

Reviewer #4: Yes

4. Have the authors made all data underlying the findings in their manuscript fully available?

Reviewer #3: Yes

Reviewer #4: No

5. Is the manuscript presented in an intelligible fashion and written in standard English?

Reviewer #3: Yes

Reviewer #4: Yes

6. Review Comments to the Author

Reviewer #3: The language in submitted article is clear, correct, and unambiguous. I did not find any more typographical or grammatical errors

Reviewer #4: The manuscript describes an important research data which may inform HIV intervention programs in similar settings. However, I would suggest minor editorial changes as described below.

Abbreviations

According to PlosOne guidelines, any abbreviation should be defined on the first appearance in the text and you should not use non-standard abbreviations unless they appear in the text at least three times.

Line 36: CAG already defined in line 34 so there was no need to repeat full definition in the subsequent sections of the abstract. I suggest you do this for all other abbreviations in the main text to ensure consistency.

Line 41: MSF not defined on its first appearance in the text and appeared only once in the abstract text.

References

Reference must follow PlosOne guidelines. Please check the reference list ad ensure PlosOne guidelines have been followed

e.g. Line 401: The source for reference number one is not shown.

7. PLOS authors have the option to publish the peer review history of their article (what does this mean?). If published, this will include your full peer review and any attached files.

Reviewer #3: No

Reviewer #4: No

---

## [Editor Report · Acceptance letter]

30 Jun 2021

PONE-D-20-06492R2 

Evaluation of a community-based HIV test and start program in a conflict affected rural area of Yambio County, South Sudan 

Dear Dr. Ferreyra:

I'm pleased to inform you that your manuscript has been deemed suitable for publication in PLOS ONE. Congratulations! Your manuscript is now with our production department. 

Kind regards, 

on behalf of

Dr. Joel Msafiri Francis 

Academic Editor

PLOS ONE